# Dynamics of IgM and IgG responses to the next generation of engineered Duffy binding protein II immunogen: Strain-specific and strain-transcending immune responses over a nine-year period

**Camila M. P. Medeiros[1,2], Eduardo U. M. Moreira[1], Camilla V. Pires[1], Letícia M. Torres[1,2], Luiz F. F. Guimarães[1], Jéssica R. S. Alves[1], Bárbara A. S. Lima[1], Cor J. F. Fontes[3], Helena L. Costa[1], Cristiana F. A. Brito[1], Tais N. Sousa[1], Francis B. Ntumngia[4], John H. Adams[4], Flora S. Kano[1]\*, Luzia H. Carvalho[1,2]\***

1 Centro de Pesquisas René Rachou/FIOCRUZ Minas, Belo Horizonte, MG, Brazil, 2 Departamento de Parasitologia, Instituto de Ciências Biológicas, Universidade Federal de Minas Gerais, Belo Horizonte, MG, Brazil, 3 Hospital Júlio Muller, Universidade Federal de Mato Grosso, Cuiabá, Mato Grosso, Brazil, 4 Center for Global Health and Infectious Diseases Research, College of Public Health, University of South Florida, Tampa, Florida, United States of America

\* luzia.carvalho@fiocruz.br (LHC); flora.kano@fiocruz.br (FSK)

**Data Availability Statement:** All relevant data are available within the article and its Supporting Information files.

## Abstract

### Background

A low proportion of *P. vivax*-exposed individuals acquire protective strain-transcending neutralizing IgG antibodies that are able to block the interaction between the Duffy binding protein II (DBPII) and its erythrocyte-specific invasion receptor. In a recent study, a novel surface-engineered DBPII-based vaccine termed DEKnull-2, whose antibody response target conserved DBPII epitopes, was able to induce broadly binding-inhibitory IgG antibodies (BIAbs) that inhibit *P. vivax* reticulocyte invasion. Toward the development of DEKnull-2 as an effective *P. vivax* blood-stage vaccine, we investigate the relationship between naturally acquired DBPII-specific IgM response and the profile of IgG antibodies/BIAbs activity over time.

### Methodology/principal findings

A nine-year follow-up study was carried-out among long-term *P. vivax*-exposed Amazonian individuals and included six cross-sectional surveys at periods of high and low malaria transmission. DBPII immune responses associated with either strain-specific (Sal1, natural DBPII variant circulating in the study area) or conserved epitopes (DEKnull-2) were monitored by conventional serology (ELISA-detected IgM and IgG antibodies), with IgG BIAbs activity evaluated by functional assays (*in vitro* inhibition of DBPII–erythrocyte binding). The results showed a tendency of IgM antibodies toward Sal1-specific response; the profile of Sal1 over DEKnull-2 was not associated with acute malaria and sustained throughout the observation period. The low malaria incidence in two consecutive years allowed us to

**Funding:** This work was supported by The National Research Council for Scientific and Technological Development-CNPq (422257/2016-8 by LHC); The Research Foundation of Minas Gerais- FAPEMIG (APQ-02625-15 by LHC); Programa Fiocruz de Fomento à Inovação: Inova Fiocruz (VPPCB-007-FIO-18-2-33 by FSK); NIH Research Project Grant Program (R01AI064478 by JHA and LHC). LHC, CFAB, TNS and CFJF are research fellows from CNPq. Scholarships were sponsored by the Coordination for the Improvement of Higher Education Personnel-CAPES (BASL, JRSA, LFFG and LMT)) and CNPq (CMPM, HLC, CVP and EUMM). We also thank the financial support from the Program for Institutional Internationalization of the Higher Education Institutions and Research Institutions of Brazil-CAPES-PrInt from FIOCRUZ and UFMG. The funders had no role in study design, data collection and analysis, decision to publish, or preparation of the manuscript.

**Competing interests:** The authors have declared that no competing interests exist.

demonstrate that variant-specific IgG (but not IgM) antibodies waned over time, which resulted in IgG skewed to the DEKnull-2 response. A persistent DBPII-specific IgM response was not associated with the presence (or absence) of broadly neutralizing IgG antibody response.

## Conclusions/significance

The current study demonstrates that long-term exposure to low and unstable levels of *P. vivax* transmission led to a sustained DBPII-specific IgM response against variant-specific epitopes, while sustained IgG responses are skewed to conserved epitopes. Further studies should investigate on the role of a stable and persistent IgM antibody response in the immune response mediated by DBPII.

## Introduction

*Plasmodium vivax* is characterized by dormant liver stage hypnozoite-parasites responsible for high frequency of relapses [1], which imposes a challenge for the current policies of malaria control and elimination. With great potential for transmission from first generation of blood-stage infection [2, 3] and lower levels of parasitemia often undetected by routine surveillance [4, 5], the proportion of malaria infections attributed to *P. vivax* has increased in areas of relatively low transmission [6].

Although there are major hurdles for vivax malaria elimination, clinical immunity to *P. vivax* is acquired much more rapidly than for *P. falciparum* [revised in [7, 8], even in low transmission settings, which make the development of an effective vaccine worth pursuing. Duffy binding protein II (DBPII) is a leading *P. vivax* malaria vaccine candidate that binds the Duffy antigen receptor for chemokines (DARC) on reticulocytes is critical for reticulocyte invasion [9, 10]. Although naturally acquired DBPII antibodies tend to be biased towards strain-specific responses [11–13], our project identified the epitope targets of protective neutralizing IgG antibody response to overlap conserved residues essential for receptor binding and DBP dimerization [12, 14–19]. Individuals able to produce these broadly binding-inhibitory antibodies (BIAb) to DBPII present reduced risk of clinical *P. vivax* malaria [20, 21]. In pursuing a structural vaccinology approach, our project created surface-engineered DBPII vaccine candidate, DEKnull-2, that retains the conserved functional epitopes needed for receptor binding and DBP dimerization but removed residues of variant nonfunctional epitopes associated with strain-specific immune responses [22]. Naturally-occurring protective immunity associated with induction of long-term memory IgG responses have anti-DBPII BIAb and DBPII reactive cells that are highly reactive with DEKnull-2.

While it is well established that naturally acquired IgG antibody responses are associated with protective clinical immunity to blood-stage malaria [23, 24], the role of IgM is not well defined [25, 26]. A recent study in a murine model of malaria demonstrated that *Plasmodium*-specific IgM memory B cells are somatically hypermutated, high-affinity, and dominate the early memory response to recurring malaria infections [27]. Likewise, the production of protective IgM antibodies during experimental malaria provides evidence of additional mechanisms by which the immune system controls *Plasmodium* infection [28, 29]. These results might explain recent data associating the depth and breadth of *Plasmodium*-specific IgM antibodies with genetic resistance to malaria infection [30], and with the reduced risk of clinical

malaria in a cohort of children [31]. Taken together, we are in accordance with other that suggest that IgM antibodies seem to be much more than just an early responder to malaria infection [32], and should be investigated during the development of vaccines.

Therefore, the goal of this study is to understand the mechanisms that underlie the broader humoral immune responses against the novel engineered DBPII vaccine candidate, DEKnull-2 [22], including the IgM response. For that, we took advantage of the long-term follow-up study previously carried out in the Amazon rainforest, where different profiles of DBPII-specific IgG responders were identified [33]. We have examined the frequency and distribution of DBPII-specific IgM and IgG antibodies during a 9-years follow-up period. As IgM antibodies may be necessary to sustain an optimal long-term protective IgG response [34–36], we also investigate whether a stable DBPII-specific IgM response could interfere with the profile of antibodies able to block the interaction ligand-receptor.

## Materials and methods

### Study area and population

The study was carried-out in the agricultural settlement of Rio Pardo (1˚46'S—1˚54'S, 60˚ 22'W—60˚10'W), Presidente Figueiredo municipality, Northeast of Amazonas State in the Brazilian Amazon region. The study site and malaria transmission patterns were described in detail elsewhere [37–39]. In this area, malaria transmission is considered hypo to mesoendemic, and the majority of residents were natives of the Amazon region [37]. Inhabitants of the settlement live on subsistence farming and fishing along the small streams. In the study area, *P. falciparum* malaria incidence has decreased drastically in recent years, and *P. vivax* is now responsible for all clinical malaria cases reported (S1 Fig).

### Study design and cross-sectional surveys

A population-based open cohort study was initiated in November of 2008, and included three cross-sectional surveys carried at six-months interval (baseline, 6 and 12-months) as previously reported [37, 39]. Briefly, (i) interviews were conducted through a structured questionnaire to obtain demographical, epidemiological, and clinical data; (ii) physical examination, including body temperature and spleen/liver size were recorded according to standard clinical protocols; (iii) venous blood was collected for individuals aged five years or older (EDTA, 5 mL), or blood spotted on filter paper (finger-prick) for those aged <5 years; and (iv) examination of Giemsa-stained thick blood smears for the presence of malaria parasites by light microscopy. The geographical location of each dwelling was recorded using a hand-held 12-channel global positioning system (GPS) (Garmin 12XL, Olathe, KS, USA) with a positional accuracy of within 15 m. Additional cross-sectional surveys were carried-out six (August 2014), seven (July 2015) and nine years later (July 2017) [22, 33]. During the long-term follow up study, the number of malaria cases fluctuated in the study area, reflecting period of high (I and III) and low (II) malaria transmission (S1 Fig). For the current study, the non-eligible criteria were (i) refusal to sign the informed consent; (ii) children, as clinical immunity is not prevalent in Amazon children [40]; (iii) pregnant women; (iv) any other morbidity that could be traced; and (v) individuals who were unable to be recruited during at least two consecutive cross-sectional surveys. The 163 participants who were eligible to the current study matched the original adult population (n = 300) for age, sex, malaria exposure [33]. Eighty-eight (54%) and 77 (47%) out of 163 eligible subjects could be recruited six and nine years later, respectively. Fifty-seven (35%) subjects presented consecutive samples throughout the 9-years follow-up period.

The ethical and methodological aspects of this study were approved by the Ethical Committee of Research on Human Beings from the René Rachou Institute (Reports No. 007/2006, No. 07/2009, No.12/2010, No. 26/2013 and CAAE 50522115.7.0000.5091), according to the Resolutions of the Brazilian Council on Health (CNS-196/96 and CNS-466/2012).

## Laboratory diagnosis of malaria

At the time of blood collection, all individuals were submitted to a finger-prick for malaria diagnosis by light microscopy. The Giemsa-stained thick blood smears were prepared and examined by experienced local microscopists, according to the malaria diagnosis guidelines of the Brazilian Ministry of Health (2009) [41]. Species-specific PCR assays targeting different plasmodial targets (*18S rRNA* gene and non-ribosomal Pvr47/Pfr364 sequences) were carried-out essentially as previously described [42]. For this, genomic DNA was extracted from either whole blood samples collected in EDTA, or from dried blood spots on filter paper using the Puregene blood core kit B (Qiagen, Minneapolis, MN, USA) or the QIAmp DNA mini kit (Qiagen), respectively, according to manufacturers' instructions.

## Recombinant blood stage *P. vivax* proteins

**DBPII-related antigens.** Recombinant DBPII-related proteins included amino acids 243–573 of the Sal-1 reference strain, DBPII-Sal1 [43], and an engineered DBPII termed DEKnull-2 [22]. These proteins were expressed as a 39kDa 6xHis fusion protein, properly refolded, as previously described [16, 22].

## Immunoglobulin (Ig) M and IgG detection assays

A conventional enzyme-linked immunosorbent assay (ELISA) for antigen-specific IgM and IgG antibody response was carried out as previously described [38], with plasma samples diluted at 1:100 (IgG) or 1:400 (IgM). Recombinant proteins were used at a final concentration of 3 μg/ml (DBPII and DEKnull-2). For each protein, the results were expressed as ELISA reactivity index (RI), calculated as the ratio of the mean optical density (OD at 492 nm) of each sample to the mean OD plus three standard deviations of samples from 20–30 unexposed volunteers. Values of RI > 1.0 were considered positive.

## Anti-DBPII erythrocyte-binding-inhibitory antibodies

The functional proprieties of DBPII antibodies (binding-inhibitory activity, BIAbs) were performed on a subset of the study population comprising 57 individuals matched for age, sex and malaria exposure. Plasma samples were tested for inhibition of DBPII-erythrocyte binding at 1:40 dilution by the standard COS7 cell assay as described [44]. A pool of *P. vivax* immune serum able to inhibit erythrocyte binding, and naïve serum was used as positive and negative control respectively. Binding was quantified by counting rosettes observed in 10–20 fields of view (x200). Percent binding-inhibition was quantified by assessing the percentage of rosettes in wells of transfected cells in the presence of test plasma relative to rosettes in wells in the presence of negative control plasma sample. Plasma samples with more than 50% inhibition of DBPII-erythrocyte binding were considered inhibitory.

## Statistical analysis

A database was created using Epidata software (http://www.epidata.dk). The graphics and the analysis were performed using GraphPad Prism version 7—GraphPad Software, La Jolla California USA, and R statistical software (version 3.3.3). Differences in proportions were

evaluated by chi-square ($X^2$) test or Fisher's exact tests, as appropriate. The Shapiro-Wilk test was performed to evaluate normality of variables. Differences in means were tested using either the one-way ANOVA, with Turkey's post hoc, or the Mann-Whitney test or Kruskal–Wallis tests, with Dunn's post hoc test, as appropriate. Linear correlation between variables, such as levels of antibodies and recent episodes of malaria, was determined by using the Spearman's correlation coefficient. Only variables associated with statistical significance at the 5% level were maintained in the final models.

## Results

### Subject characteristics and antibody profiles to DBPII-related antigens at enrollment

The median age of individuals included in the study was 42 years (IQR: 28–53) with a 1.1:1 proportion of male to female (Table 1). The age was significantly associated with a subject's time of malaria exposure in the Amazon area (r = 0.75; p<0.0001, Spearman's correlation test). At the time of their first blood collection, the overall prevalence of malaria was 13%, with all infections caused exclusively by *P. vivax*; 3% of infections were detected by conventional microscopy, and 10% by a species-specific Real-Time PCR.

To determining whether DBPII antibody responses included both strain specific and broadly reactive antibodies, two different DBPII allelic variants were tested, Sal1, a common variant circulating in the study area and the DBPII reference strain, and the engineered DEKnull-2 whose antibody response target conserved DBPII epitopes. While 39% of the individuals enrolled in the study had IgM antibodies to DBPII-Sal1, only 23% had detectable IgM antibodies to the conserved DEKnull-2 ($X^2$ = 9.698 p = 0.018). Considering IgG antibodies, a similar proportion of individuals (46–43%) responded to each of recombinant protein assayed (p>0.05).

**Table 1. Demographic, epidemiological and immunological characteristics of 163 individuals at enrolment.**

| Characteristics | | |
|---|---|---|
| Median age, years (IQR) | | 42 (28–53) |
| Gender, male: female | | 1.1: 1 |
| Previous malaria self-reported episodes, median (IQR) | | 5 (3–15) |
| Years of residence in Amazon area, median (IQR) | | 35 (24–50) |
| Location of residence in Amazon area, riverine: non-riverine | | 1: 1 |
| Acute *P. vivax* infection: | | |
| | Patent *P. vivax* infection[1], *n* (%) | 5 (3) |
| | Sub patent *P. vivax* infection[2], *n* (%) | 17 (10) |
| | Total, n (%) | 22 (13) |
| Antibody response[3], positive *n* (%): | | |
| **IgM**[*] | DBPII-Sal1 | 64 (39) |
| | DEKnull-2 | 37 (23) |
| **IgG** | DBPII-Sal1 | 75 (46) |
| | DEKnull-2 | 69 (43) |

IQR = InterQuartile Range.

[1]Positive *P. vivax* infections detected by conventional light microscopy.

[2]Positive *P. vivax* infections detected by real-time PCR.

[3]Evaluated by conventional ELISA serology using recombinant proteins against *P. vivax* Duffy binding protein region II (DBPII).

[*]Statistically different (Chi Square test with Yates correction, $X^2$ = 9.698; p = 0.018).

## Composition and dynamics of strain-specific and strain-transcending DBPII antibody repertoire over a nine-year period

**IgM DBPII-related response.**   Over three cross-sectional surveys at 6-month intervals (high transmission, Phase I), between 40% and 37% of individuals had DBPII-Sal1 IgM antibodies, as detected by conventional serology (S2 Fig). Broadly reactive antibodies (DEKnull-2) were detected at significantly lower frequencies (24–16%). During all follow-up period DBPII-Sal1 IgM antibodies predominate over DEKnull-2 antibodies, and it was independent of the levels of malaria transmission in the study area (phase I, II and III). The levels of antibodies (evaluated here by medians of reactivity) showed a similar tendency towards strain-specific IgM antibodies (S2 Fig and S1 Table).

**IgG DBPII-related response.**   The frequencies and levels of IgG antibody response to both DBPII proteins were similar over the first 12-months period (phase I) (S3 Fig). Despite of that, antibody levels varied among responders, including individuals with strong IgG antibody response (RI > 10) to both DBPII-related antigens. Of interest, the intensity of malaria transmission influenced in the proportion of Sal-1 versus DEKnull-2 immune responses. In the low transmission period (phase II), a significant decrease in strain-specific IgG antibodies was observed while the frequencies and levels of DEKnull-2 remained similar to the baseline (47% *vs*. 45–39%) (S3 Fig and S1 Table). The profile of DEKnull-2 over Sal1-specific IgG response was maintained until the end of the study (phase III, high transmission).

**Ratio IgG to IgM antibodies.**   The ratio IgG/IgM to each recombinant protein confirmed that IgG but not IgM Sal1-antibodies were sensitive to malaria transmission intensity (Fig 1). Anti-Sal1 IgG antibodies decreased during a transmission period, and this profile remained until the end of the study. The low malaria transmission period strengthened DEKnull-2 antibodies, especially for IgG antibodies (Fig 1B).

## Influence of acute infection in the DBPII antibody repertoire

During the 9-years follow-up study period, 42 (26%) out of 163 studied individuals had a detectable *P. vivax* infection, most of them detected only during the first 12-month period (n = 36, Phase I). Although this subgroup was not differentiated from the study population by age (42 vs. 39 yrs-old), gender (1.3:1 vs. 1.1:1), or time of malaria-exposure (34 vs. 33 yrs), the majority of them were classified as riverine population (30 out of 42, 71%).

In these long-term exposed individuals, there was a predominance of sub patent (PCR-positive) over patent (microscopy-positive) *P. vivax* infections, and acute infection was not associated with the presence of either IgM or IgG antibody responses to any of the proteins, i.e., Sal1 or DEKnull-2 (Fig 2). Specifically, individuals with persistent malaria infection (for example, RP553 and RP555) did not present DBPII-related IgM antibodies, and vice-versa, a persistent IgM antibody response to both proteins (RP416 and RP516) was not associated with a potential booster by blood-stage infections. The absence of correlation between infection and antibody response was also observed for IgG antibodies (RP405, RP416 and RP433). This data is confirmed by the Spearman's correlation coefficient between IgG and IgM antibody responses (S4 Fig).

## IgG and IgM antibodies repertoire according to the immunological background

Plasma samples were screened for anti-DBPII BIAbs to investigate the relationship between IgM/ IgG antibody responses and the profile of BIAbs responders' DBPII inhibitory immune responses. These were classified as persistent non-responders (NR) characterized by the

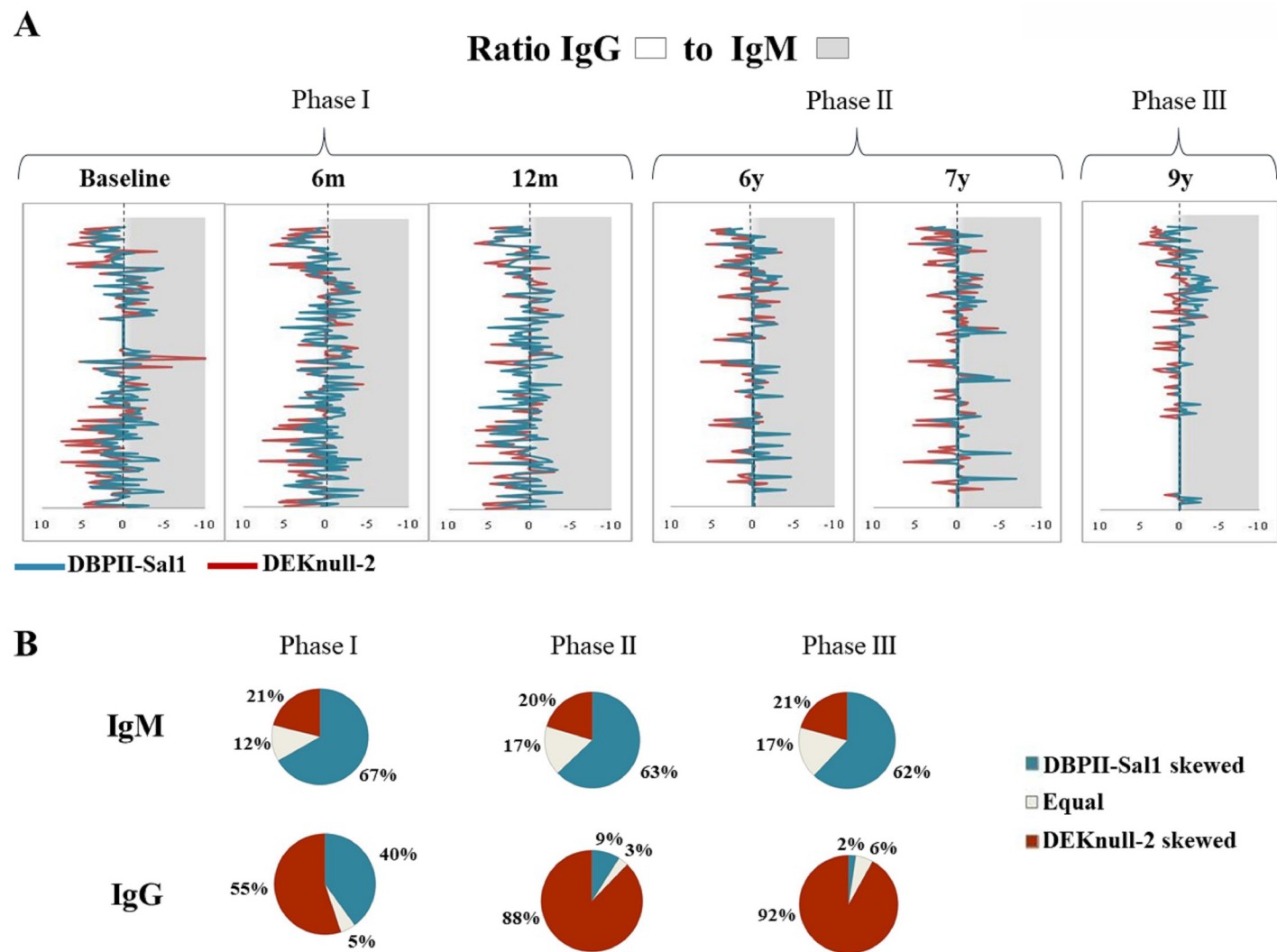

**Fig 1. Ratio of IgG to IgM antibody responses against DBPII-Sal1 and DEKnull-2 during the 9 years follow-up study period. (A)** The ratio between IgG and IgM response against DBPII-Sal1 (blue line) and DEKnull-2 (red line) for all subjects enrolled in the study was represented for each cross-sectional survey. The longitudinal study comprised six cross-sectional surveys, which included periods of high (phase I and III) and low (phase II) malaria transmission; the first three cross-sectional surveys were carried-out during the first year (baseline, 6 and 12 months), and three carried-out 6[th], 7[th] and 9[th] years later. Individual responses towards IgG were represented on left (white) and towards IgM on right (grey). Individuals with equal IgG and IgM responses were show on central dotted line. **(B)** Pie charts representing the percentage of subjects with the following IgG and IgM response profiles: DBPII-Sal1 skewed, equal, and DEKnull-2 skewed during the three phases of transmission.

absence of BIAbs antibodies; temporary responders (TR) whose BIAbs response alternated between positive/negative, and persistent responders (PR) whose BIAbs were stable throughout the study (Fig 3). During the low transmission period (Phase II), IgG responses were consistently detected against both recombinant proteins (DBPII-Sal1 and DEKnull-2) for the persistent BIAbs responders and, less with the temporary BIAbs responders. All but one (RP416) of PR subgroup (93%, n = 15) had DEKnull-2 IgG antibodies, while only 7 out of 13 (54%) of TR reacted to DEKnull-2. As expected, the majority of NR did not have detectable long-term IgG. IgM antibody responses were not associated with any profile of BIAbs responders, with some individuals with broadly and long-term IgM responses detected in all subgroups (Fig 3).

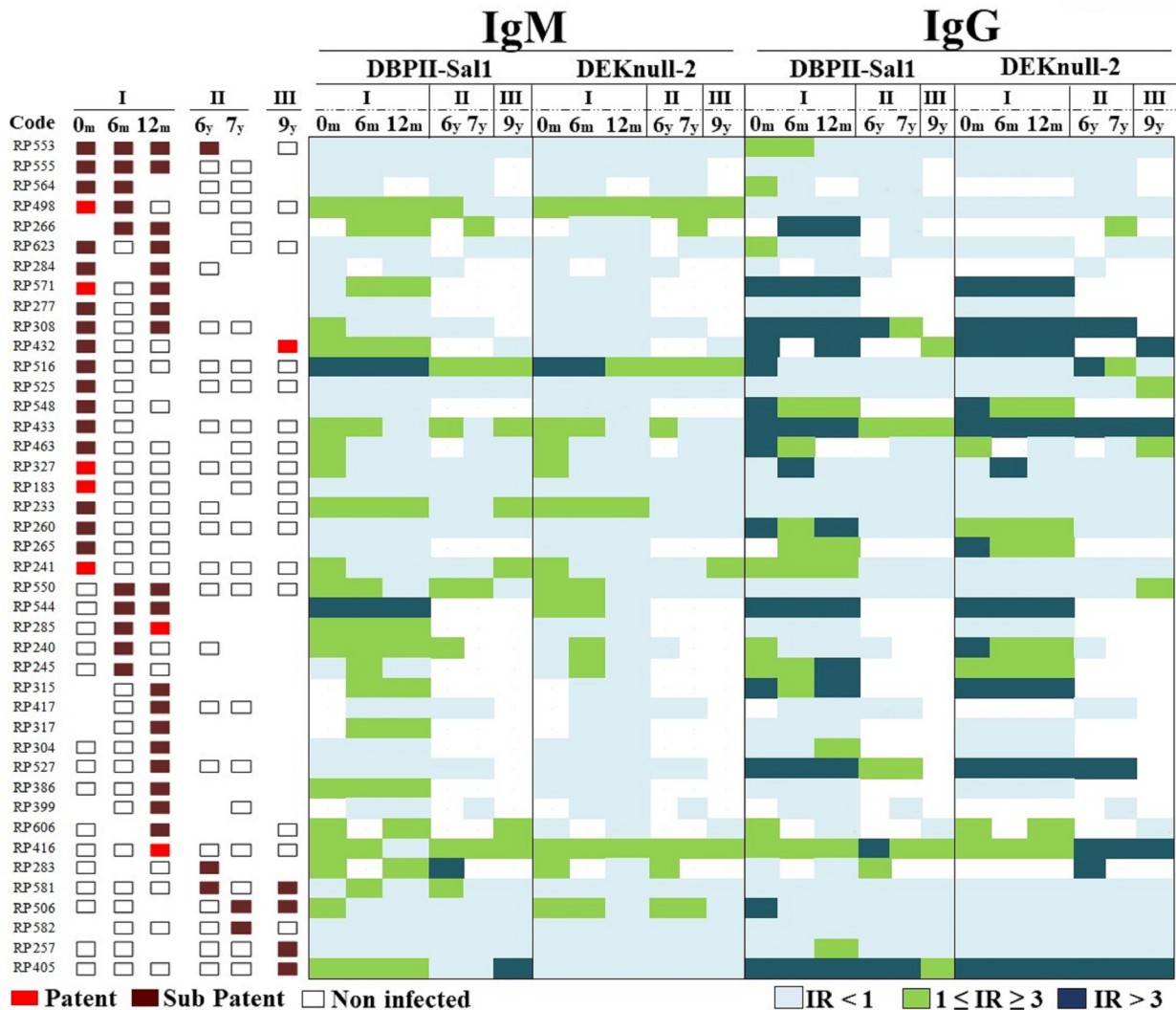

**Fig 2. IgM and IgG antibody responses to DBPII-Sal1 and DEKnull-2 during *P. vivax* infections.** Each line represents one subject who had detectable *P. vivax* blood stage infection at any time of the follow-up study. The longitudinal study comprised six cross-sectional surveys, which included periods of high (I and III) and low (II) malaria transmission; the first three cross-sectional surveys were carried-out during the first year (zero, 6 and 12 months; 0-12m), and three carried-out 6th, 7th and 9th years later (6-9y). Forty-two subjects were positive for *P. vivax* infections and classified into *sub patent* and *patent* infection accordingly to PCR or conventional microscopy diagnosis, respectively (red color variation). In the right panel, the green color variation in the heatmap shows IgM and IgG responses for each cross-sectional survey. ELISA antibody responses were expressed as Reactivity Index (RI) calculated by dividing the mean optical density (OD at 492 nm) of each sample to the mean OD plus three standard deviations of samples from unexposed volunteers. Values of RI > 1.0 were considered positive (on a green scale).

## Discussion

In an Amazonian community exposed to low levels intermittent malaria transmission, we sought to investigate the relationship between anti-DBPII IgM and IgG antibody responses reactive with strain-specific (Sal1) or strain-transcending (DEKnull-2) immune responses [22]. In general, the frequencies and levels of IgM antibodies showed a tendency towards strain-specific antibodies (Sal1). Interestingly, the response profile of Sal1 over DEKnull-2 IgM antibodies was sustained throughout the 9-years observation period, including in consecutive years in which malaria transmission dropped drastically in the study area (Phase II).

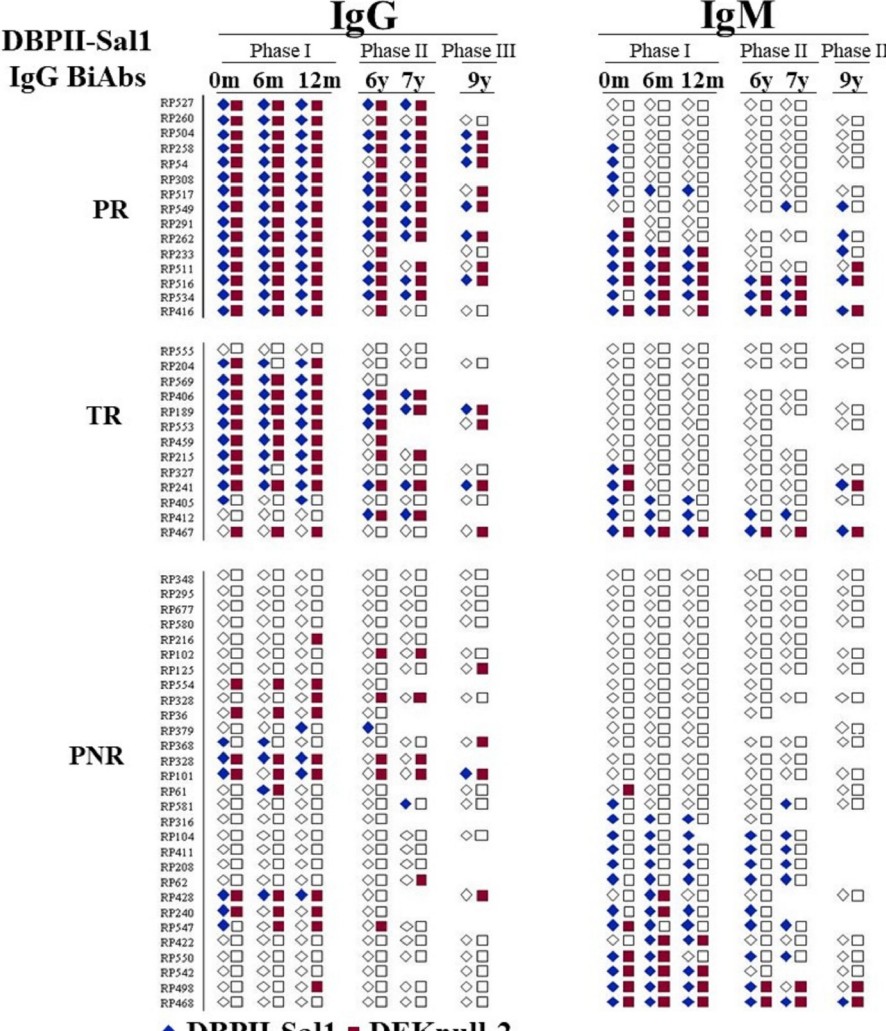

**Fig 3. Profile of IgM and IgG antibody responses of Rio Pardo subjects previously classified according to the DBPII Binding Inhibitory Antibodies (BIAbs).** According to their BIAbs response, malaria-exposed individuals (n = 57) were previously characterized as [22]: i) *Persistent responder (PR)*, who had BIAbs response during the 9 years of follow-up; ii) *Temporary responders (TR)*, who had variable BIAbs response during those cross-sectional surveys and iii) *Persistent non-responders (NR)*, who had no BIAbs detected any time in the study. Each line represents individual IgM and IgG antibody responses against DBPII-Sal1 or DEKnull-2, in each cross-sectional survey. Colored symbols indicate positive antibody response at ELISA (IR > 1) and white symbols negative response at ELISA (IR ≤ 1). ELISA results were expressed as Reactivity Index (IR), with Reactivity Index (RI) > 1 considered positive.

Although the majority of high IgM responses (RI> 3) were no longer detected in the low transmission period, the results suggest that individuals in *P. vivax*-endemic Amazonian communities were able to sustain their DBPII-specific IgM antibody responses. These results may explain recent findings showing that IgM-expressing memory B cells are expanded in malaria patients living in endemic areas [27], and confirm data from other studies, which suggest that IgM antibodies may play an underappreciated role in immune response against malaria infections [29–31]. Although scant longitudinal data are available about secondary IgM responses in *P. vivax*-exposed populations, a recent prospective study undertaken in a low transmission area of Western India demonstrated that *P. vivax* alters peripheral B-cell profiles and induces

parasite-specific IgM that persisted post recovery [45]. While the results from India are in concordance with a persistent IgM response, antigen-specific antibody responses were evaluated only over a short period-of time, i.e., during acute infection and upon 30 days post-treatment. Here, we demonstrated that long-term *P. vivax* exposure to low and unstable levels of malaria transmission can lead to a sustained DBPII-specific IgM response. At this time, it is not possible to define if this persistent DBPII-related IgM response could indicate that IgM-experienced B cells needs to be constantly activated or if this response is rather associated with a bona-fide memory [29, 46]. Of relevance, *P. falciparum*-specific IgM antibodies were detected for more than 6 months in Australians returning from malaria endemic areas [31]. Together, the findings of long-term IgM response emphasize the need to understand the role of the IgM specific antibodies in both natural infection and vaccine antigens.

Our results showing an absence of correlation between antigen-specific IgM antibodies and acute malaria infection are intriguing, considering that in response to infection the antibodies made initially are usually IgM confined to the intravascular pool [26, 47]. Although unknown, we hypothesize that antigenically distinct DBPII variants could be responsible to new blood-stage infections, including relapses. Still, this hypothesis may not explain many of the acute infections since Sal1 was the most prevalent DBPII variant circulating at that time in the study area [37]. As sub patent infections predominate in the study area (77% by PCR vs. 23% by microscopy), one may speculate that these low parasite densities were not sufficient to provide a booster of IgM antibody response. However, recent findings in experimental malaria infections demonstrated that secondary IgM response was not affected by either challenge dose or the time of rechallenge [27]. Perhaps a more plausible and not mutually exclusive explanation for why IgM antibodies were not associated with peripheral blood stage infections is the relative underrepresentation of *P. vivax* asexual stages in patient blood [48, 49]. Nowadays, a considerable body of evidence indicates a tissue reservoir, such as bone marrow, where most *P. vivax* parasite burden resides [50]. Consequently, we cannot exclude that *P. vivax*-specific IgM production is dependent upon tissue parasite persistence, a phenomenon that needs to be investigate. In different experimental models, low-level of chronic infection may provide sufficient antigen to maintain IgM plasmablasts in the bone marrow either through inflammation [51] or antigenic stimulation [52].

From high to low malaria incidence in two consecutive years of the study was critical to elucidate the relative contribution of IgG versus IgM antibodies to DBPII immune response. Variant-specific IgG but not IgM antibodies waned over time, which resulted in IgG skewed reactivity to the DEKnull-2 antigens lacking the variant epitopes but retaining the conserved epitopes. This profile of IgG response was not unexpected as we previously demonstrated that a significant number of long-term malaria-exposed individuals mount a strong and stable IgG response toward conserved DEKnull-2 epitopes [22, 33]. Of further interest, the levels of IgM antibodies were not related to IgG antibodies, and the lack of correlation occurred to both DBPII-related antigens. It has been proposed that the correlation (or not) between IgG/IgM in malaria may reflect inherent structural differences between antigens, including the relative conservation of the epitopes that are targeted; for example, strong correlations between IgG and IgM was observed to MSP2, whereas this was not seen with MSP1-19 and AMA-1 [53]. In the case of *P. vivax* MSP1, antibody response to different regions of the protein appeared distinct, with the N-terminal portion predominantly associated with IgM antibodies and C-terminal with IgG response [54–57]. In the case of DBPII, several studies have mapped functional immunoreactive B cell epitopes associated with broadly neutralizing IgG antibody response [12, 14, 15, 18], but no data is available about IgM response.

It has been shown that IgM antibodies may be necessary to sustain an optimal long-term protective IgG response [34–36]. Assuming that functionally acquired IgG antibodies able to

broadly inhibit DBPII-DARC interaction (BIAbs) are associated with a reduced risk of clinical *P. vivax* malaria [20, 21], we sought to investigate whether a stable DBPII-specific IgM response could interfere with the profile of BIAbs responder. It is particularly relevant as we demonstrated that long-term DEKnull-2 responders with high levels of IgG antibodies are able to produce a persistent BIAbs response [22]. Stratification of the DBPII BIAbs responders (persistent, temporary and no-responder) showed no association (positive or negative) between antigen-specific IgM response and the profile of BIAbs. Additional investigation may help define the contribution (if any) of anti-DBPII IgM antibodies in the immune response mediated by *P. vivax*.

The present study has limitations that must be considered when interpreting the results. As DBPII sequences from all cross-section surveys were not available, antibody response against a DBPII variant circulating in the study area (Sal1) was used to characterize species-specific immune response. Our previous studies confirm Sal1 as major local DBPII variant [37, 58, 59], whose antibody response is highly prevalent in Amazonian exposed-individuals [33, 38]. Consequently, we are confident that Sal1 is a key DBPII variable for assessing the species-specific immune response in the study area. Taken together, our results demonstrated that IgG (but not IgM) variant-specific DBPII antibodies were poorly sustained at low transmission period, which confirms that IgM antibodies may be more indicative of continuous exposure to malaria, whereas epitope-conserved IgG antibodies are relatively stable and associated with BIAbs response. The reason for the persistence of the IgM response in our cohort merits further investigations.

## Supporting information

**S1 Fig. Temporal distribution of malaria cases in the agricultural settlement of Rio Pardo (Amazonas, Brazil) during 9 years follow-up study.** *P. vivax* (blue) and *P. falciparum* (red) microscopy diagnosed case report data in Rio Pardo were provided by the National Malaria Surveillance Information System (SIVEP-Malaria) and plotted per month. The longitudinal study comprises six cross-sectional surveys during 2008–2017, which includes periods of high (dark-grey, phase I and III) and low (light-grey, phase II) malaria transmission; the first three cross-sectional surveys were carried-out during the first year (baseline, 6 and 12 months); three carried-out 6[th], 7[th] and 9[th] years later. Modified from Pires *et al.*, 2018 [33]. (TIF)

**S2 Fig. The levels of IgM antibody response against DPBII-Sal1 and DEKnull-2 during the study period.** The IgM responses were expressed as Reactivity Index (RI), with Reactivity Index (RI)>1.0 considered positive. The individual values are represented by blue (DBPII--Sal1) and red (DEKnull-2) open circles. Transversal lines indicate medians and interquartile ranges. The cross-sectional surveys were carried-out as described in legend to S1 Fig. The frequency of seropositive subjects (Pos (%)) on each cross-sectional survey is represented below each graphic. Different number of asterisks indicate the variation on p value (*p< 0.05 to **p<0.0001; Fisher's exact test), for significance differences between the frequency of DBPII--Sal1 and DEKnull-2 response. (TIF)

**S3 Fig. The levels of antibodies IgG response against DPBII-Sal1 and DEKnull-2 during the study period.** The IgG responses were expressed as Reactivity Index (RI), with Reactivity Index (RI)>1.0 considered positive. The individual values are represented by blue (DBPII--Sal1) and red (DEKnull-2) open circles. Transversal lines indicate medians and interquartile ranges. The cross-sectional surveys were carried-out as described in legend to S1 Fig, with IgG

original data obtained from Pires et al., 2018 [33]. The frequency of seropositive subjects (Pos (%)) on each cross-sectional survey is represented below each graphic. Different number of asterisks indicates the variation on p value ($^*$p< 0.05 to $^{**}$p<0.0001; Fisher's exact test), for significance differences between the frequency of DBPII-Sal1 and DEKnull-2 response.
(TIF)

**S4 Fig. Spearman correlation between IgM and IgG antibody responses against DBPII-Sal1 (red) and DEKnull-2 (blue) of subjects with or without acute *P. vivax* infections.** The correlation between IgM and IgG antibodies response were performed separately to subjects with acute *P. vivax* infections (closed triangle) and non-infected individuals (open circles) to each protein.
(TIF)

**S1 Table. Levels of IgM and IgG antibodies response against *P. vivax* DBPII-proteins during the 9 years follow-up study.**
(DOCX)

## Acknowledgments

We thank the inhabitants of Rio Pardo for enthusiastic participation in the study; the local malaria control team in Presidente Fiqueiredo for their logistic support; the units of Fundação Oswaldo Cruz in Manaus, AM (Fiocruz Amazonia), and Belo Horizonte, MG (Fiocruz Minas), for overall support.

## Author Contributions

**Conceptualization:** Francis B. Ntumngia, John H. Adams, Flora S. Kano, Luzia H. Carvalho.

**Data curation:** Camila M. P. Medeiros, Flora S. Kano, Luzia H. Carvalho.

**Formal analysis:** Camila M. P. Medeiros, Camilla V. Pires, Letícia M. Torres, Cor J. F. Fontes, Cristiana F. A. Brito, Flora S. Kano, Luzia H. Carvalho.

**Funding acquisition:** John H. Adams, Flora S. Kano, Luzia H. Carvalho.

**Investigation:** Camila M. P. Medeiros, Camilla V. Pires, Letícia M. Torres, Tais N. Sousa, Francis B. Ntumngia, John H. Adams, Flora S. Kano, Luzia H. Carvalho.

**Methodology:** Camila M. P. Medeiros, Eduardo U. M. Moreira, Camilla V. Pires, Letícia M. Torres, Luiz F. F. Guimarães, Jéssica R. S. Alves, Bárbara A. S. Lima, Cor J. F. Fontes, Helena L. Costa, Cristiana F. A. Brito, Tais N. Sousa, Francis B. Ntumngia, Flora S. Kano, Luzia H. Carvalho.

**Project administration:** Flora S. Kano, Luzia H. Carvalho.

**Resources:** Cor J. F. Fontes, Cristiana F. A. Brito, Tais N. Sousa, John H. Adams, Flora S. Kano, Luzia H. Carvalho.

**Supervision:** Flora S. Kano, Luzia H. Carvalho.

**Validation:** Camila M. P. Medeiros, Camilla V. Pires, Letícia M. Torres, Luiz F. F. Guimarães, Jéssica R. S. Alves, Bárbara A. S. Lima, Cor J. F. Fontes, Helena L. Costa, Tais N. Sousa, Francis B. Ntumngia, Flora S. Kano, Luzia H. Carvalho.

**Visualization:** Camila M. P. Medeiros, Camilla V. Pires, Letícia M. Torres, Flora S. Kano.

**Writing – original draft:** Camila M. P. Medeiros, Camilla V. Pires, Letícia M. Torres, Flora S. Kano, Luzia H. Carvalho.

**Writing – review & editing:** Camila M. P. Medeiros, Eduardo U. M. Moreira, Camilla V. Pires, Letícia M. Torres, Luiz F. F. Guimarães, Jéssica R. S. Alves, Bárbara A. S. Lima, Cor J. F. Fontes, Helena L. Costa, Cristiana F. A. Brito, Tais N. Sousa, Francis B. Ntumngia, John H. Adams, Flora S. Kano, Luzia H. Carvalho.

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
