## [Decision Letter · Decision Letter 0]

26 Mar 2020

PONE-D-20-02118

Dynamics of IgM and IgG responses to the next generation of engineered Duffy binding protein II immunogen: strain-specific and strain-transcending immune responses over a nine-year period

PLOS ONE

Dear Dr. Carvalho,

Thank you for submitting your manuscript to PLOS ONE. After careful consideration, we feel that it has merit but does not fully meet PLOS ONE’s publication criteria as it currently stands. Therefore, we invite you to submit a revised version of the manuscript that addresses the points raised during the review process.

Based on the careful assessment form the expert reviewers in this field, this manuscript will be satisfactory after the careful revision of the description pointed by them. Please take all the suggestions into consideration for your revision. I really appreciate your patience.

We would appreciate receiving your revised manuscript by May 10 2020 11:59PM. To enhance the reproducibility of your results, we recommend that if applicable you deposit your laboratory protocols in protocols.io, where a protocol can be assigned its own identifier (DOI) such that it can be cited independently in the future. For instructions see: http://journals.plos.org/plosone/s/submission-guidelines#loc-laboratory-protocols

We look forward to receiving your revised manuscript.

Kind regards,

Takafumi Tsuboi

Academic Editor

PLOS ONE

Journal Requirements:

Please ensure that your manuscript meets PLOS ONE's style requirements, including those for file naming. The PLOS ONE style templates can be found at http://www.plosone.org/attachments/PLOSOne_formatting_sample_main_body.pdf and http://www.plosone.org/attachments/PLOSOne_formatting_sample_title_authors_affiliations.pdf

Reviewers' comments:

Reviewer's Responses to Questions

**Comments to the Author**

1. Is the manuscript technically sound, and do the data support the conclusions?

Reviewer #1: Yes

Reviewer #2: Partly

2. Has the statistical analysis been performed appropriately and rigorously? 

Reviewer #1: Yes

Reviewer #2: Yes

3. Have the authors made all data underlying the findings in their manuscript fully available?

Reviewer #1: Yes

Reviewer #2: No

4. Is the manuscript presented in an intelligible fashion and written in standard English?

Reviewer #1: Yes

Reviewer #2: Yes

5. Review Comments to the Author

Reviewer #1: Medeiros et al. have studied the dynamics of IgM and IgG responses against P. vivax Duffy binding protein region II (PvDBPII) in cross-sectional surveys in a P. vivax endemic region in the Brazilian Amazon. They have used recombinant PvDBPII SalI and PvDBPII DEKnull-2 proteins for both ELISA and binding inhibition assays with endemic sera to study development of strain-specific and strain-transcending antibodies against PvDBPII respectively. The authors demonstrate for the first time the presence of long-lasting IgM antibodies directed against PvDBPII, which recognize PvDBPII SalI better than PvDBPII DEKnull-2 and also inhibit RBC binding to PvDBPII SalI better than PvDBPII DEKnull-2. In contrast IgG show the reverse specificity suggesting IgGs more commonly recognize strain-transcending epitopes in PvDBPII. This is the first study to systematically analyse IgM and IgG against PvDBPII and although the mechanism or reasons for these differences in specificity are not known and the implications for protective immunity are not understood, the descriptive analysis presented here is interesting and important. It emphasizes the importance and need to study both IgM and IgG responses against parasite antigens that are leading vaccine candidates. The study is technically sound and the analysis provides interesting insights into development of antibody responses against PvDBPII following natural exposure to P. vivax.

The authors have responded to the queries raised earlier adequately.

Reviewer #2: This manuscript presents to investigate the detail of IgM & IgG responses against DBPII-Sal1 & DEKnull-2 during the cross-sectional surveys study over nine-year period in the Amazon region. The authors argue that long term exposure to low levels of P. vivax transmission led to a sustained variant-specific IgM responses (DBPII-Sal1 over DEKnull-2), while sustained IgG responses are skewed to conserved epitopes (DEKnull-2 over DBPII-Sal1). Overall, this manuscript is of interest, clearly written some points, informative and statistical analysis has been properly done.

However, some of these findings had been reported using same field study from same group. And then it is required the authors to present these data with clearly and carful editing to avoid misleading as a redundant publication. The major IgG response against DBPII-Sal1 & DEKnull-2 was reported on Ref 35, fig 3. And then it may consider the major finding on this manuscript is IgM response against these antigens during long term field survey. Although the major purpose of this study has not been fully justified yet, especially it is still unclear the focus on IgM responses against DBPII-Sal1 & DEKnull-2 such a long term survey. And also a significant problem of this manuscript is figure 1 has been used on previous article, and it should be removed from main MS or move it on to supplemental figure.

Careful editing will greatly strengthen the manuscript, making it more clear and concise. Specific comments are as follows:

1. The authors has to justify the major purpose of this study, why focus on IgM response against DBPII-Sal1 & DEKnull-2 during such a long 9 year period? In generally, IgM antibodies appear early in the course of an infection and usually reappear, to a lesser extent, after further exposure. And it often found to bind to specific antigens, even in the absence of prior immunization. And also it was reported IgM as a rapid and early responder in Malaria (Ref 27). And also it is not justified yet to select DBPII-Sal1 & DEKnull-2 antigens for this study due to the authors mentioned these sentence as below (Ref 35) “Thus, while the DEKnull-2 antibody response is not a useful tool for serological evaluation of malaria transmission, it seems to be appropriate to evaluate naturally-acquired immunity to P. vivax blood-stage infections. At the individual level, DEKnull-2 immune responses confirmed the highest acquired immunity of HR group as compared with IR or LR group; all HR subjects were positive to DEKnull-2, and remained seroreactive until the end of the 9 year follow-up study. Despite of that, the levels of antibodies reacting to DEKnull-2 declined during the study, including in the high transmission phase III.”

Why did authors focus on IgG and IgM responses against DBPII-Sal1 and DEKnull-2? And what is the major new finding on the current study?

2. It is required the authors that ref 35 has to refer in the introduction on this manuscript, especially it is very important to mention it what they found it before. I assume this MS is a kind of retrospective serological follow up study, due to this, it has to be refereed well all the detail for them previous work. And also they need to explain the advantage of BIAbs assay due to clarify for the difference from Ref 35.

3. Fig 1 had been used on them previous article (Ref 35, fig1), and then it has to remove from main MS or move it on to supplemental figure. (Such a minor change will not be called as a “modified”)

4. Fig 2 and 3 have to remove from main MS or move it on to supplemental figure. These figures are just raw data sets for fig 4.

5. The information of right panel of fig 5 is not founded on the current manuscript. What is this meaning? What is the IR? The reviewer would not be founded any advantage points about this figure except left part. Moreover it is useful to combine the Pv-infected information (fig 5 left panel) with figure 6. Taken together, it would be getting better to merge fig 5 and 6.

6. PLOS authors have the option to publish the peer review history of their article (what does this mean?). If published, this will include your full peer review and any attached files.

Reviewer #1: No

Reviewer #2: No

---

## [Author Response · Author response to Decision Letter 0]

2 Apr 2020

RESPONSES TO REVIEWERS

REVIEWER # 1 

Major comment. (…) The authors demonstrate for the first time the presence of long-lasting IgM antibodies directed against PvDBPII (…) This is the first study to systematically analyze IgM and IgG against PvDBPII (…)the descriptive analysis presented here is interesting and important. (… )the analysis provides interesting insights into development of antibody responses against PvDBPII following natural exposure to P. vivax. The authors have responded to the queries raised earlier adequately.

AUTHORS: We thank the reviewer for the careful examination of the manuscript and rebuttal letter, and to consider our work well carried-out and relevant for this field of investigation. 

REVIEWER # 2 

Major comment 

Rev#2 (…) Overall, this manuscript is of interest, clearly written some points, informative and statistical analysis has been properly done (…). However, some of these findings had been reported using same field study from same group. And then it is required the authors to present these data with clearly and carful editing to avoid misleading as a redundant publication (…). The major IgG response against DBPII-Sal1 & DEKnull-2 was reported on Ref 35, fig 3. 

AUTHORS: First of all, we thank the reviewer for the careful examination of our manuscript and consider the results relevant for this field of investigation. Regarding the concern of redundant publication, we must clarify that the MS from Pires et al (ref 35) was design to evaluate the seroconversion rates as a tool for a rapid assessment of P.vivax malaria transmission intensity. For that, we include blood-stage antigens covering a wide range of immunogenicity such as MSP1-19, AMA-1 and DBPII-related proteins. An unsupervised learning algorithm such as k-means clustering allowed categorizing individuals based on the magnitude and breadth (specificities for multiple antigens including AMA-1, DBPs and MSP-1) of their conventional antibody response. In that MS (ref 35), we demonstrate that IgG antibodies against AMA-1/MSP1-19 were much more appropriate to detect temporal fluctuation of P.vivax transmission that the less immunogenic DBPII-related proteins. Consequently, in the previous MS we investigated which antigen should be most appropriate for detecting fluctuations in P.vivax malaria transmission.

Aiming towards universal strain-transcending DBPII immunity, in the current MS, we evaluate for the first time the long-term DBPII-specific IgM response and the relationship of IgM antibodies with (i) conventional IgG response and their (ii) ability to blocking the interaction ligand-receptor (binding inhibitory activity/BIAbs, as detected by COS-7 assays). Consequently, there is no overlap between the current MS and ref 35. As scant longitudinal data are available about secondary IgM responses in P. vivax-exposed populations, we are confident that our findings fill a gap in the literature for providing the first description of the of the dynamic of IgM versus IgG responses to a novel surface-engineered P.vivax DBPII-based vaccine (the relevance of studying IgM antibodies was included below; Major comment/Specific comment 1). 

Finally, we would like to emphasize that long-term follow-up studies are expensive, time consumer, and offer an exceptional opportunity to investigate unappreciated topics. Consequently, to focus on DBPII-specific IgM responses, we took advantage of the long-term follow-up study previously carried out in the Amazon rainforest, where different profiles of DBPII-specific IgG responders were identified (as described in ref 35). We are confident that the results presented here will contribute to further optimize the design of new DBPII immunogens such as DEKnull-2. 

In the revised version of the MS, we clarified this concern raised by the reviewer (please see track changes copy, for example, in the last paragraph of introduction). Also we explained in the legend of Fig 3 (now S3 Fig, as requested by the reviewer) that IgG original data were obtained from Pires et al., 2018 (previous ref 35, and in the revised version ref 33).

Rev#2: Major comment -Rev#2: (…)it is still unclear the focus on IgM responses against DBPII-Sal1 & DEKnull-2 such a long term survey. (…)

Specific comment -1 1. The authors have to justify (…), why focus on IgM response against DBPII-Sal1 & DEKnull-2 during such a long 9 year period? (…)Why did authors focus on IgG and IgM responses against DBPII-Sal1 and DEKnull-2? And what is the major new finding on the current study? (…) In generally, IgM antibodies appear early in the course of an infection and usually reappear, to a lesser extent, after further exposure.(…) And also it is not justified yet to select DBPII-Sal1 & DEKnull-2 antigens for this study due to the authors mentioned these sentence as below (Ref 35) “Thus, while the DEKnull-2 antibody response is not a useful tool for serological evaluation of malaria transmission, it seems to be appropriate to evaluate naturally-acquired immunity

AUTHORS: Currently, a significant body of evidence point to an underappreciated role for IgM in protection against infectious disease including malaria; for example: 

(1) In experimental malaria, mice lacking specific-IgG antibodies but with strong anti-parasite IgM antibody responses develop long-lasting non-sterile immunity and survive lethal parasite challenge (Borges da Silva et al., 2018; PMID: 30148845); 

(2) Somatically hypermutated Plasmodium-specific IgM+ MBCs proliferated and gave rise to antibody-secreting cells that dominated the early secondary response to parasite rechallenge (Krishnamurty et al., 2016; PMID:27473412); 

(3) IgM antibodies (but in much less extension IgG) to a broad array of P. falciparum antigens was associated to genetic resistance to malaria (Arama et al., 2015; PMID: 26361633);

 (4) In P.falciparum malaria, merozoite-specific IgM is an important functional and long-lived antibody response targeting blood-stage malaria parasites that contributes to malaria immunity (Boyle et al., 2019; PMID: 31579826);

(5) IgM antibodies may be necessary to sustain an optimal long-term protective IgG response (Harte et al., 1983; PMID: 6835362; Boes et al., 1998; PMID: 9590224),

Taken together, we are in accordance with other that suggest that IgM antibodies seem to be much more than just an early responder to malaria infection (Boonyaratanakornkit & Taylor, 2019; PMID: 31603585), and should be investigated during the development of vaccines. 

In pursuing a P.vivax DBPII structural vaccinology approach, we created DEKnull-2 antigens lacking the variant epitopes but retaining the conserved protective functional epitopes, which are essential for binding its cognate reticulocyte receptor. As quoted above, DEKnull-2 should be used to evaluate natural acquired immunity and not fluctuation of P.vivax transmission (ref. 35). Considering the new paradigm of the relevance of IgM response in malaria (as summarized above), we sought to investigate here the relationship between the IgM and IgG antibody responses against conserved (DEKnull-2) and variable DBPII epitopes circulating in the area. In the current study, a comparison between conserved and variable epitopes was critical, as DBPII-based vaccine should target strain-transcending immune responses. As IgM antibodies may be necessary to sustain an optimal long-term protective IgG response (PMID: 6835362; PMID: 9590224), we also investigate whether a stable DBPII-specific IgM response could interfere with the profile of functional antibodies, i.e., antibodies able to block the interaction ligand-receptor (BIAbs). 

Here, we demonstrated for the first time that long-term exposure to low and unstable levels of P. vivax transmission lead to a sustained DBPII-specific IgM response toward variant-specific epitopes while sustained IgG responses are skewed to conserved epitopes involved in protection (DEKnull-2 over Sal-1). Although a persistent DBPII-specific IgM response was not associated with broadly neutralizing IgG antibody response, future studies should investigate on the role of IgM in the DBPII acquired immune response (perhaps to sustain IgG response). We are confident that these findings will contribute to the current stage of development of DBP-based vaccines that focus broad protective immune response. 

In the revised version of the MS we clarify this topic, as requested to the reviewer (see MS with track changes). More specifically, although most of the references on the importance of studying IgM antibodies were already included in the penultimate paragraph of the introduction, we reinforced this topic in the final paragraphs of the introduction and included additional references.

Rev#2: And also a significant problem of this manuscript is figure 1 has been used on previous article, and it should be removed from main MS or move it on to supplemental figure.

AUTHORS: the original Fig 1 was included in the Methods to clarify the study design and the long-term follow up cohort that was carried before (ref35); otherwise, the readers will need to access the ref 35. However, as requested to reviewer, Fig. 1 was included in the revised version of the MS as supplementary (now Fig 1S).

Rev#2: Careful editing will greatly strengthen the manuscript.

AUTHORS: The MS was revised by Dr. John H Adams, South Florida University, USA (https://health.usf.edu/publichealth/overviewcoph/faculty/john-adams). Dr. Adams is also a co-author of the MS. 

Specific point 2. It is required the authors that ref 35 has to refer in the introduction on this manuscript, especially it is very important to mention it what they found it before. I assume this MS is a kind of retrospective serological follow up study, due to this, it has to be refereed well all the detail for them previous work. And also they need to explain the advantage of BIAbs assay due to clarify for the difference from Ref 35.

AUTHORS: As we clarified above (please see major comments) the MS from Pires et al (ref 35) was design to evaluate seroconversion rates as a tool for assess trends in P.vivax malaria transmission; in that study to evaluate the breadth of antibody response was critical to included antigens covering a wide range of immunogenicity (MSP1-19, AMA-1 and DBPIIs). On the other hand, the current MS was design to get insides in the acquired immune response to a novel engineered vaccine candidate - DEKnull-2. As quoted before, we have included in the last paragraph of the introduction that we took advantage of the long-term follow-up study previously carried out in the Amazon rainforest, where different profiles of DBPII-specific IgG responders were identified (Pires et al., 2018). In the current MS, it was critical to evaluate BIABs response because a strong naturally acquired BIAb response is associated with a reduced risk of clinical P. vivax malaria (King et al., 2008; PMID: 18523022; Nicolete et al., 2016; PMID: 27578850). All these references were included in the MS (please see highlighted text in the 2nd paragraph of the introduction, track changes MS). 

Specific point 3. Fig 1 had been used on them previous article (Ref 35, fig1), and then it has to remove from main MS or move it on to supplemental figure. (Such a minor change will not be called as a “modified”)

AUTHORS: Okay, it was done. In the revised version of the MS, Fig 1 was included as supplementary Fig S1

Specific point 4. Fig 2 and 3 have to remove from main MS or move it on to supplemental figure. These figures are just raw data sets for fig 4.

AUTHORS: Okay, it was done. Figures were adjusted as suggested (Fig S2 and S3)

Specific point 5. The information of right panel of fig 5 is not founded on the current manuscript. (…)

AUTHORS: Perhaps the results in Fig. 5 have led to a misunderstanding. More specifically, Fig 5 shows every single individual who had blood-stage infection at any time of the follow-up study (left panel), and in the right panel (the heatmap) their correspondent antibody responses to DBPII-related antigens, including both IgM (1st part of the right panel, i.e., the first part of the heatmap) and IgG antibodies (2nd part of heatmap). At the top of heat map, IgM and IgG antibodies were identified in bold. The results showed no correlation between acute infection and either antigen-specific IgM or IgG antibodies. As example, in the text, we have identified some specific individuals (by code) and comments about their results in the MS. At this time, we should clarify that all data from the original Fig. 5 (now Fig. 2 as requested by the reviewer) were included in the text and/or in the legend of the figure (please see highlighted text in the track changes copy). 

Rev#2: (…) What is the IR? (…)

In the figure, the results of antibody response were expressed as the ELISA reactivity index (IR) that was calculated as described in the methods (please see highlighted text in the item IgM and IgG detection assays). Briefly, the ELISA reactivity (RI) was calculated as the ratio of the mean optical density (OD at 492 nm) of each sample to the mean OD plus three standard deviations of samples from 20-30 unexposed volunteers. Values of RI > 1.0 were considered positive (on a green scale). In the revised version of the MS, we included RI definition in the legend of Fig 2 (ex-Fig 5).

Rev#2:The reviewer would not be founded any advantage points about this figure except left part. 

AUTHORS: We are confident about the relevance of the data presented in the figure 5, particularly, due to the absence of correlation between antigen-specific IgM antibodies and acute malaria infection. These findings are intriguing as the antibodies made initially against infection are usually IgM. Perhaps a plausible explanation is the relative underrepresentation of P. vivax asexual stages in patient blood as a considerable body of evidence indicates a tissue reservoir, such as bone marrow, where most P. vivax parasite burden resides (please see 2nd paragraph of discussion). Consequently, we decided to keep figure 5 (now retitled as Fig. 2).

Rev #2: Moreover it is useful to combine the Pv-infected information (fig 5 left panel) with figure 6. Taken together, it would be getting better to merge fig 5 and 6.

AUTHORS: While the figure 5 shows the absence of correction between antigen-specific antibody responses and acute malaria infection, Fig 6 shows the IgM/IgG antibody responses according to the profile of binding inhibitory activity (BIAbs) as detected by functional assays (COS-7 transfected cells). Consequently, it is not possible to merge results that are unrelated. 

We would like to thank deeply the reviewer for his time and effort, and we hope that the MS is now suitable for publication.

---

## [Decision Letter · Decision Letter 1]

22 Apr 2020

Dynamics of IgM and IgG responses to the next generation of engineered Duffy binding protein II immunogen: strain-specific and strain-transcending immune responses over a nine-year period

PONE-D-20-02118R1

Dear Dr. Carvalho,

We are pleased to inform you that your manuscript has been judged scientifically suitable for publication and will be formally accepted for publication once it complies with all outstanding technical requirements.

With kind regards,

Takafumi Tsuboi

Academic Editor

PLOS ONE

Additional Editor Comments (optional):

Reviewers' comments:

Reviewer's Responses to Questions

**Comments to the Author**

1. If the authors have adequately addressed your comments raised in a previous round of review and you feel that this manuscript is now acceptable for publication, you may indicate that here to bypass the “Comments to the Author” section, enter your conflict of interest statement in the “Confidential to Editor” section, and submit your "Accept" recommendation.

Reviewer #2: All comments have been addressed

2. Is the manuscript technically sound, and do the data support the conclusions?

Reviewer #2: Yes

3. Has the statistical analysis been performed appropriately and rigorously? 

Reviewer #2: Yes

4. Have the authors made all data underlying the findings in their manuscript fully available?

Reviewer #2: Yes

5. Is the manuscript presented in an intelligible fashion and written in standard English?

Reviewer #2: Yes

6. Review Comments to the Author

Reviewer #2: The authors have been addressed all reviewer's comments.

The current MS is suitable for publication.

7. PLOS authors have the option to publish the peer review history of their article (what does this mean?). If published, this will include your full peer review and any attached files.

Reviewer #2: No

---

## [Editor Report · Acceptance letter]

24 Apr 2020

PONE-D-20-02118R1 

Dynamics of IgM and IgG responses to the next generation of engineered Duffy binding protein II immunogen: strain-specific and strain-transcending immune responses over a nine-year period 

Dear Dr. Carvalho:

I am pleased to inform you that your manuscript has been deemed suitable for publication in PLOS ONE. Congratulations! Your manuscript is now with our production department. 

With kind regards,

on behalf of

Prof. Takafumi Tsuboi 

Academic Editor

PLOS ONE